# A Deep Trajectory Controller for a Mechanical Linear Stage Using Digital Twin Concept

Kantawatchr Chaiprabha and Ratchatin Chancharoen *

Department of Mechanical Engineering, Faculty of Engineering, Chulalongkorn University, Bangkok 10330, Thailand
* Correspondence: ratchatin.c@chula.ac.th; Tel.: +6682-218-6643

**Abstract:** An industrial linear stage is a device that is commonly used in robotics. To be precise, an industrial linear stage is an electro-mechanical system that includes a motor, electronics, flexible coupling, gear, ball screw, and precision linear bearing. A tight fit can provide better precision but also generates a difficult-to-model friction that is highly nonlinear and asymmetrical. Herein, this paper proposes an advanced trajectory controller based on a digital twin framework incorporated with artificial intelligence (AI), which can effectively control a precision linear stage. This framework offers several advantages: detection of abnormalities, estimation of performance, and selective control over any situation. The digital twin is developed via Matlab's Simscape and runs concurrently having a real-time controller.

**Keywords:** motion control; trajectory following controller; digital twins; bond graph; anomaly detection; adaptive controller





## 1. Introduction

Motion control is a fundamental part in robotics that deals with the design and control of the movement of robots, machines, and other mechanical systems [1]. The development of linear motion is highly intriguing and is favored in many industrial applications [2]. Among motion control devices, an industrial linear stage is widely used in machines to produce precise linear motion in various applications, including manufacturing [3–7], testing and inspection [7–9], and material handling [10,11]. In a typical design, a linear stage is driven by various motors: a stepper motor [8], or brushed motor [3,7,11], ball screw [5] and precision linear bearing [12]. A precision linear bearing has a rigid structure. All components are precision made such that the position of a carriage can be sensed and controlled with an encoder at the rear end of the motor. This design offers high load capacity [6], accuracy [4,6,13], repeatability, and speed [14].

In recent years, computing power has dramatically increased having a smaller footprint. A machine is connected to cloud services [15,16]. Both actuation and sensing have also advanced [17,18], leading to a digital twin framework [19] that uses a virtual model for a variety of purposes such as design [20–22], simulation [23–25], analysis [26,27], and monitoring [28–30]. Similar to model predictive control [31,32], which is one approach for this framework [31,33,34], a digital twin simulates a physical plant in simulation time where internal states and outcomes can be projected [20,32,35]. For a complex system, this is very practical. Moreover, the comparison between digital and physical twin can be used to monitor the health of the physical twin or anomaly detection [36,37]. While many studies have explored the use of digital twin technology for trajectory planning and optimization [38–45], the majority of them have focused on machines and cells with multi-degrees-of-freedom motion [38–45]. Thus, a joint level digital twin is yet to be explored. In this work, the benefits of a digital twin for a trajectory controller at joint level have been studied for generating linear motion.

In this project, a new class of trajectory controller is proposed based on a digital twin system. Advances in technology mentioned earlier make it possible to construct a digital model of a physical linear stage that can be run in real-time. This leads to the next era of motion control in the way that an anomaly can be sensed [46]. Artificial intelligence can be incorporated with digital twin to practically demonstrate its power. In addition, a shuffle and adaptive controller can be selected to suit a situation so that we may tradeoff the positioning precision with something else, e.g., energy consumption, lifespan, smoothness, and safety [47]. This work focuses on: (A) a proposed trajectory controller based on a digital twin concept, and its successful implementation, (B) detection of anomalies, and (C) controller adaptability to suit the situation.

## 2. Theoretical Background

Figure 1, an investigation plant is an electro-mechanical linear stage, which is driven by a 15 W permanent magnet DC (PMDC) motor. The linear stage is installed having a 2 mm pitch lead screw and precision linear bearing. The purpose of a lead screw is to convert the rotational motion of a motor to linear translation [12,13,48]. The PMDC motor is powered by a VNH5019 H-bridge motor driver. The motor is coupled with a 500 PPR encoder to measure the shaft displacement. Thus, the carriage's displacement can be determined via the kinematic relationship of the screw mechanism.

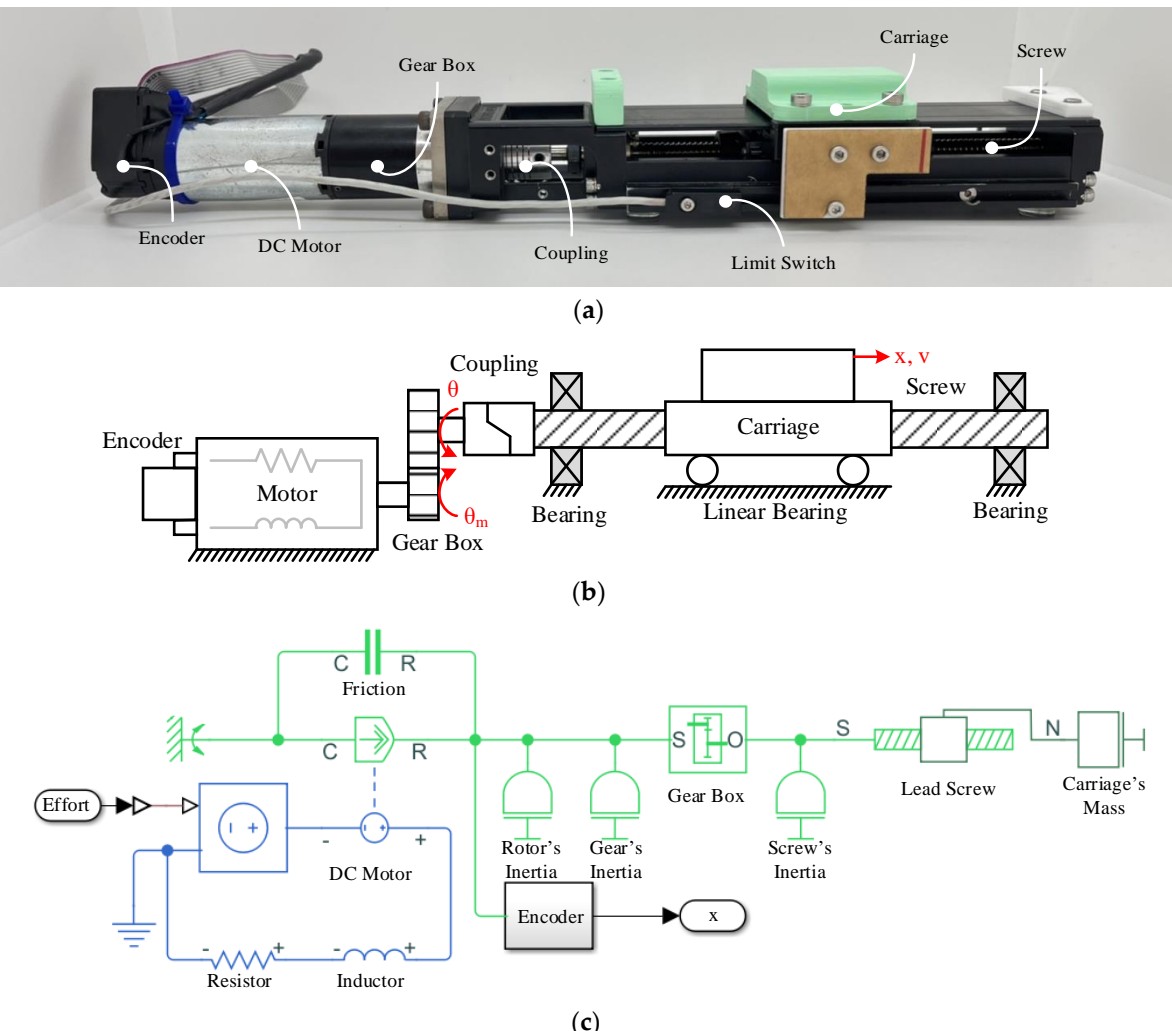

**Figure 1.** Investigation of the linear stage: (**a**) Electro−mechanical linear stage (Photograph), (**b**) Schematic diagram of the linear stage, (**c**) Simscape model.

In this conventional linear stage design, both the gear and screw mechanisms are used to provide a boost to the PMDC motor and the overfitted linear bearing is used to ensure precise linear motion. In this design, the PMDC motor can drive the carriage at near-zero back-drivability. Disturbance at the carriage side is suppressed and the closed loop control of the motor is eased. Moreover, the gear, screw, and bearing comes with a significant amount of viscous friction that drains out the kinetic energy into heat during the motion [49]. Such friction is good for the robustness and stability of the closed loop control system in most applications.

In the investigation plant, the MAXON 236662 motor is a graphite brushed cylindrical motor having an alnico magnet. The gear is MAXON 166160 planetary gear with 21:1 mechanical advantage. The drive circuit is VHN5019: 24 V that accepts a pulse-width modulation (PWM) command up to 20 kHz. In the linear stage, Misumi LX26 is equipped with a lead screw having 2 mm pitch and a diameter of 8 mm. Such a linear stage can efficiently and effectively drive the linear motion whereas the PWM commands the drive circuit. The drive circuit is highly correlated to the dynamical motion of the carriage on the linear stage.

The dynamical simulation of the electro-mechanical system has considerably advanced over recent years. For instance, advances in solving algorithms have taken place [50,51]. Both computing power and massive data handling have been significantly enhanced. These developments support a realistic and accurate complex multi-domain simulation. Parallel computing makes it possible to perform complex dynamical simulation that may be used in real-time applications.

In this work, Matlab's Simscape is employed as in block diagrams [52] and bond graphs [49,53]. The linear dynamical system can be written in *S*-domain where its input-output relation is written algebraically. A framework of block diagrams is an effective way to handle dynamical devices in a complex system [54–57] in a graphical environment [58]. A single ordinary differential equation (ODE) is set for an entire simulation but can easily be modified. Matlab's Simulink is a block diagram framework with hundreds of predefined blocks and effective working environments. The bond graph modelling framework is complementarily used with block diagrams [55,59]. The difference is that bond graphs use both across and through variables to connect blocks that effectively represent each physical device and transmission signals can flow in both directions in the model. In this way, we can model a complex system in an abstraction layer. The mathematics of each device is packed within the block and are thus easy to manage [55–57,60].

With advances in simulation, the digital twin of the electro-mechanical linear stage can be modeled and real-time simulated. In this framework, an electrical domain is shown in blue. Mechanical rotational and translational domains are represented by light green and dark green, respectively. The connection between the components is via an energy flow pipe whereby energy can flow bi-directionally. Thus, this framework can easily be applied and adapted to several scenarios within a single simulation model. Moreover, this framework creates an abstraction in which detailed parameters of each component provide a better focus on how physical components are connected and cooperate.

## 3. Design and Construction of a Dynamical Digital Twin

### 3.1. Mathematics

In the electro-mechanical linear stage, the variables (a function of time) are listed in Table 1.

**Table 1.** State variables in the electro-mechanical linear stage.

| Variable | Description | Unit |
|---|---|---|
| $V(t)$ | Voltage applied across motor | V |
| $i_a(t)$ | Current flow in motor | A |
| $T(t)$ | Motor's torque | Nm |
| $T_f(t)$ | Friction torque | Nm |
| $P(t)$ | Load force | N |
| $\theta(t)$ | Screw angle | rad |
| $x(t)$ | Carriage displacement | m |

In this system, a controllable voltage source applied voltage $V(t)$ across a DC brushed motor. Applying Kirchhoff's voltage law to the DC brushed motor model with gear yields:

$$V(t) = L\frac{\partial}{\partial t}i_a(t) + Ri_a(t) + K_b K_g \frac{d}{dt}\theta(t). \tag{1}$$

where the motor's current $i_a(t)$ flows through the motor's coil. The current $i_a(t)$ generates the torque output via the Lorentz force where the torque $T(t)$ of the motor is proportional to the current $i_a(t)$, accordingly:

$$T(t) = K_a K_g\, i_a(t). \tag{2}$$

Considering the kinematic coupling between the motor's shaft and lead screw's carriage, the motion is governed by:

$$x(t) = r_s \tan(\lambda)\,\theta(t) \tag{3}$$

where the rotational motion converts to translation motion by the screw mechanism [61].

The dynamics of the screw mechanism can be realized by applying Newton's second law on the Wedge model. The equation is shown in rotational form, as in Equation (4):

$$\left(J + m\,r_s{}^2\,\xi\,\tan(\lambda)\right)\frac{d^2}{dt^2}\theta(t) = T(t) - T_f(t) - r_s\,\xi\,P(t) \tag{4}$$

where the $\xi$ is the efficiency of ball screw [61].

*3.2. Identification of Parameters*

Table 2, theparameters in the mentioned dynamical equations are identified either from data sheets of the hardware or from the experimental system. Herein, the overall friction's behavior is complex and thus determined by experimentation (Figure 2a) [62–64].

**Table 2.** Parameters in the linear stage.

| Parameter | Description | Quantity |
|---|---|---|
| L | Motor's inductance | 0.556 mH |
| R | Motor's resistance | 0.399 Ohm |
| $K_b$ | Motor's velocity constant | 0.03525 V/(rad/s) |
| $K_a$ | Motor's torque constant | 0.03525 Nm/A |
| $K_g$ | Gear ratio | 21 |
| $J$ | Moment of inertia | 46.1 g/cm$^2$ |
| $\lambda$ | Screw angle | 0.0794 rad |
| $m$ | Carriage's mass | 1.37 kg |
| $r_s$ | Shaft radius | 4 mm |
| $\xi$ | Screw's efficiency | 0.79 |
| $\mu_s$ | Coulomb friction coefficient | 0.02 |

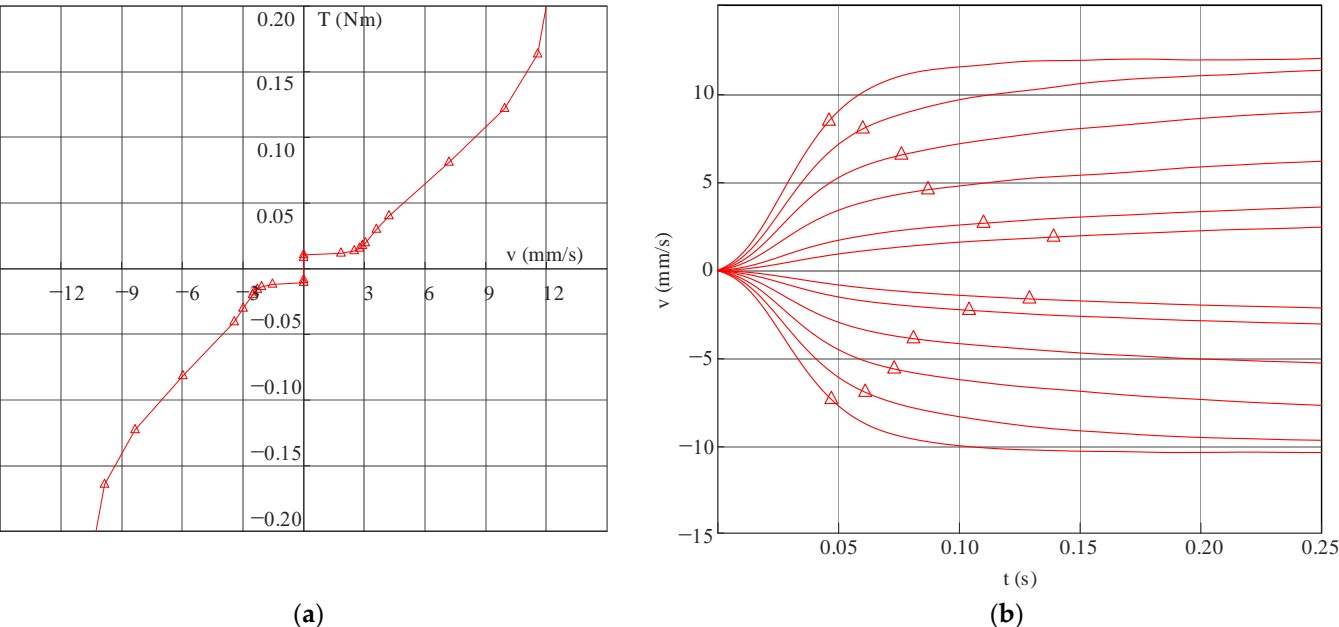

**Figure 2.** (**a**) Friction model of the system. (**b**) Velocity of carriage at 24.0, 19.2, 14.4, 9.6, 4.8, 2.4, −2.4, −4.8, −9.6, −14.4, −19.2, −24 V applied to the PMDC motor, from top to bottom. The triangles showing 70% of the terminal velocity indicate the speed response of the system.

The process of identification regards the friction model was designed to apply constant voltage to the motor. Voltage was set at a low voltage to determine the breakaway friction behavior. The position of the linear stage was acquired through the 500 PPR motor's encoder with 1 kHz sampling frequency. After that, the time series of positions was fitted via smoothing-spline, and then the velocity was estimated.

Combining Equations (1) and (2), motor torque can be determined as in Equation (5):

$$T(t) = K_a K_g \frac{\left(V(t) - K_b K_g \frac{d}{dt}\theta(t)\right)}{R}. \tag{5}$$

where $i_a(t)$ assumes to be constant. Therefore, the inductance term can be negligible.

In Figure 2a, the graph shows the relationship between velocity and motor torque, as calculated in Equation (5). Static friction torque can be determined using the data when the system remains stationary; later, low voltage is applied continuously. At a velocity between −2.5 and 2.5 mm/s, the torque due to friction remains the same. However, beyond this interval, viscous friction was dominant as the lubricant layer between the bearing surfaces formed. This viscous interval was observed linearly until maximum voltage in both directions was reached.

In Figure 2b, open-loop transient behavior of the linear stage is shown. In the time between 0 and 0.02 s, the graph illustrates the increase in acceleration as it corresponds to the acceleration of the carriage due to the open-loop voltage reference, as seen in the trend of growth in velocity. As the system gains more velocity, the viscous friction builds up and resists the effort as a result of the decrease in acceleration, which can be clearly seen by the triangle marks. The maximum velocity that this system can reach is around 12.2 mm/s and −10.3 mm/s. Thus, the existence of asymmetric friction is confirmed.

The model of the physical electro-mechanical linear stage is used to investigate predicted behavior when there are parameter changes and/or disturbances. Various controller designs can be simulated to predict the resulting outcome before a suitable one is chosen to run the physical plant.

## 4. A Typical Trajectory Controller

The linear stage has to be controlled such that its carriage moves to the desired position. The controller is closed-loop; the position of the carriage is monitored in real-time and used as feedback to locate the desired position. The difference, called an error, is fed into the controller to generate a command to actuate the linear stage in such a way that the carriage arrives at the desired position. A stiff PD controller is used in this investigation. However, since the friction in the investigation plant is considerably high, it plays as a natural derivative gain.

### 4.1. Regulatory Control

The investigation plant is experimentally driven with an electromechanical linear stage having no load. The stage is commanded to go to 1 mm position from zero position. The high gain proportional controller (K = 0.1) computes the effort beyond the hardware limit (24 V). In this period, the controller is likely to be open-loop with maximum capacity and its behavior is demonstrated in Figure 2b. It is noted that the settling time of the open loop velocity control is between 0.05–0.15 s.

In Figure 3, the maximum velocity of the linear stage in this combination is 11 mm/s. The velocity ramped up from stationary to its maximum capacity in 0.15 s (Figure 3b). Once the carriage is close to the target position where the computed control effort is less than the maximum capacity (24 V), the controlled system demonstrated the second-order dynamical system (Figure 3b from 0.08 s). In this case, an overshot of 16.25%, a settling time of 0.24 s, and a near zero steady state error are achieved. In most applications, the industrial linear stage proves to be good.

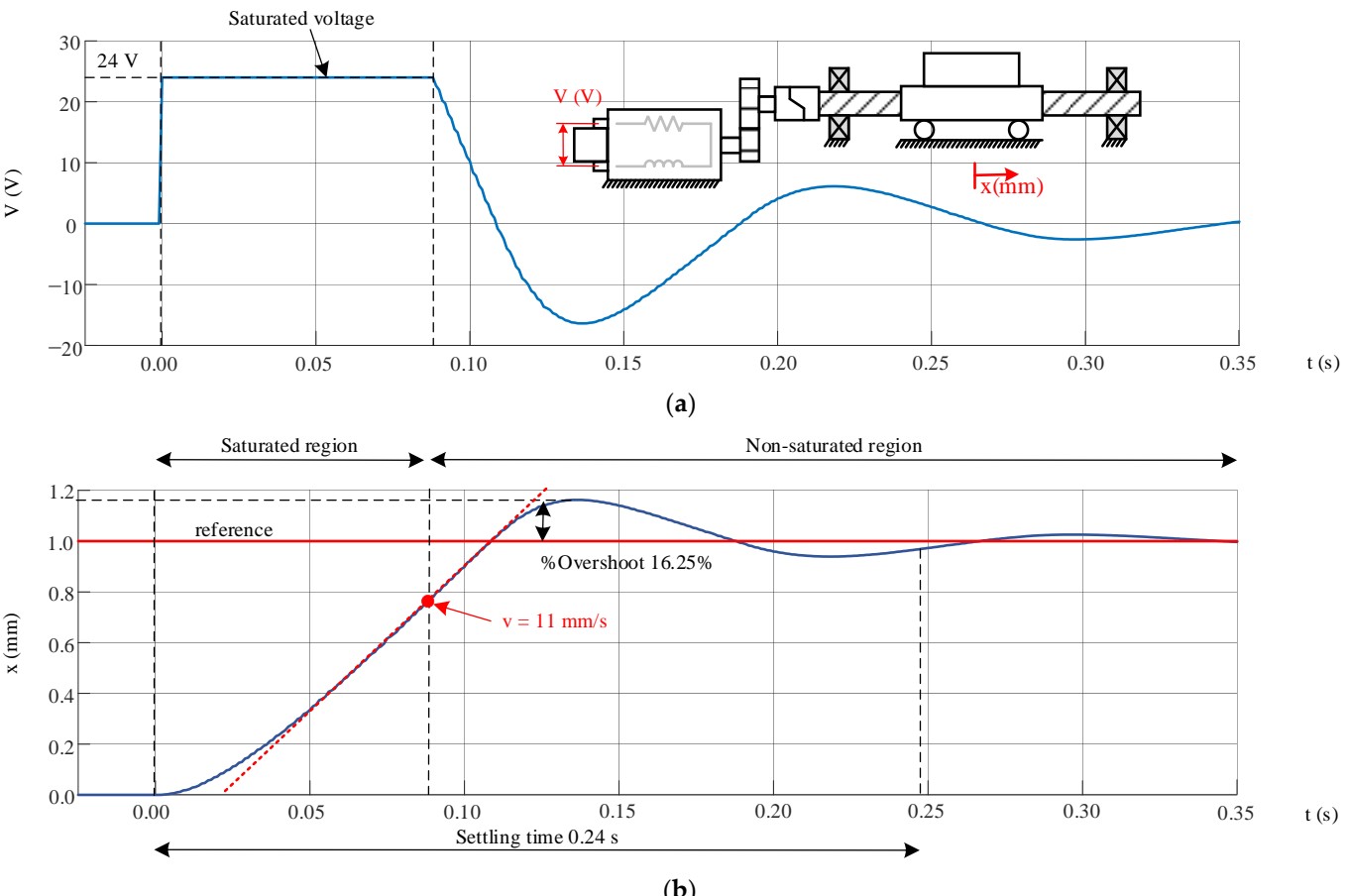

**Figure 3.** Closed−loop step responses: (**a**) Voltage input from the 24 V power supply, and (**b**) Carriage position corresponding to the input from the controller.

## 4.2. Trajectory Control

Trajectory control is more challenging since we are faced with uncertainties, disturbances, and constraints on the input. In this work, the linear stage was commanded to follow a sine sweep trajectory. The reference trajectory is sinusoidal and sweeps from 0.5 Hz to 2 Hz in 25 s. During low frequency (Figure 4 Zone I), the actual trajectory perfectly tracks the reference. However, fluctuation is noted, especially when the carriage changes its direction of motion. When frequency is high (Figure 4 Zone II), fluctuation lessens while tracking error rises. If the frequency is high (Figure 4 Zone III), fluctuation disappears but the tracking error increases along with the actuation effort.

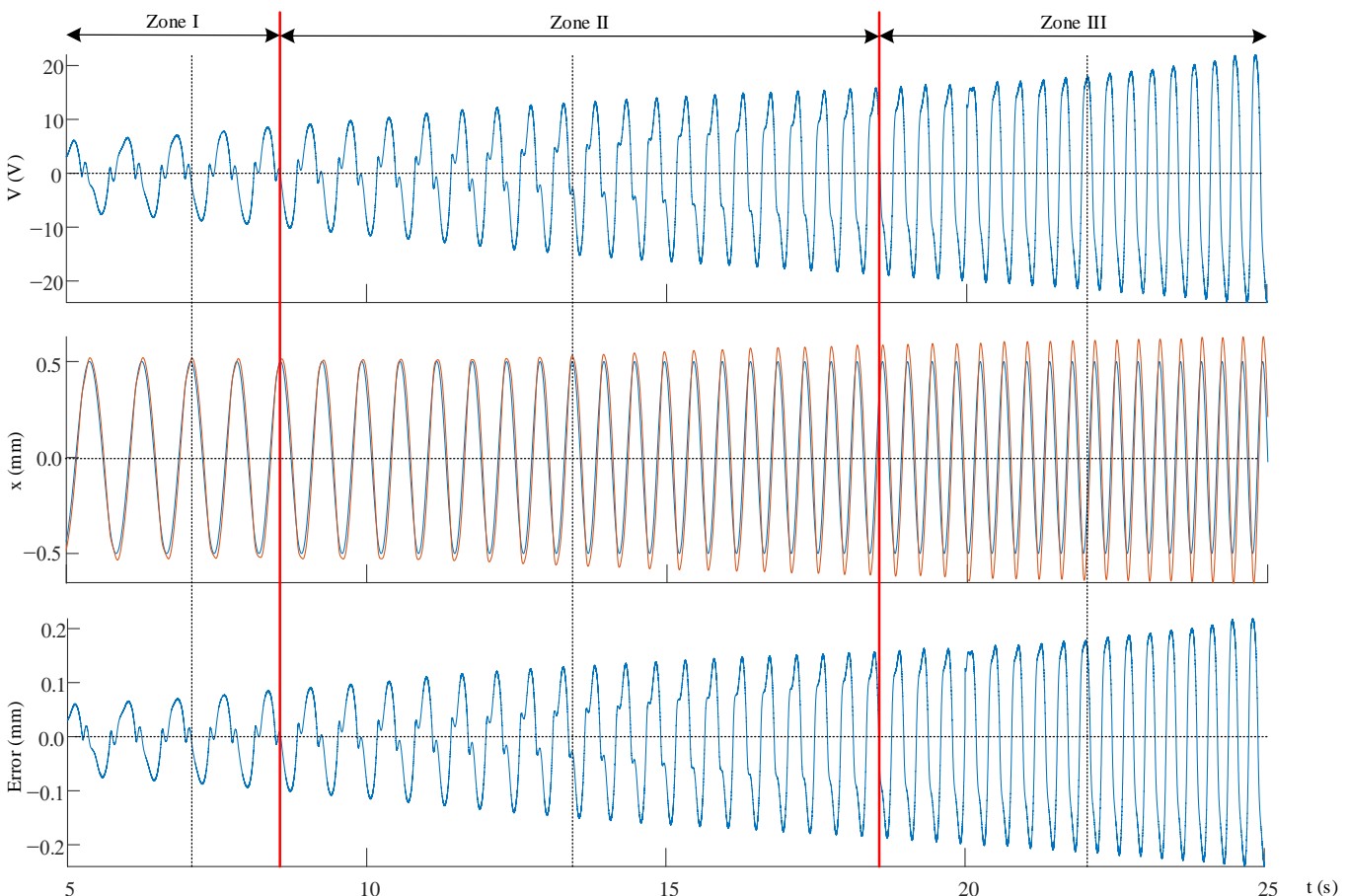

**Figure 4.** Closed−loop trajectory: experimental results.

## 4.3. Effect of the Controller Gain on the Soft/Stiff Behavior

Our trajectory controller is the stiff PD controller. The stiff controller attempts to follow a trajectory while facing uncertainties, disturbances, and nonlinear behavior. Fluctuation can occur during the trajectory. The reference trajectory is sinusoidal at 1 Hz. In Figure 5a, the effect of the soft/stiff controller's behavior is demonstrated. There is a tradeoff between the tracking error and the positioning fluctuations due to the stiffness of the controller. The resulting behavior is further analyzed. The closed-loop control system is approximated as a linear system and its input–output data pairs are used to find the dominant zero(s)/root(s) in Figure 5b. This locus plot clearly explains the behavior related to the stiff controller [52]. If the controller is stiff, the roots stay on the left side (on the real axis). When stiffness is lessened, the roots walk into the right (slower). However, if it is too soft, two roots split and go into the imaginary zone. In a real application, precision positioning or tracking is not needed all the time during the mission. For instance, if we command a linear stage to go to its home position, at its fastest speed, to reset the zero position, open

loop velocity control where the command effort is at its full capacity is suitable. If there is uncertainty or high disturbance, a soft controller may be more suitable. A soft controller is also good in terms of energy consumption and service life.

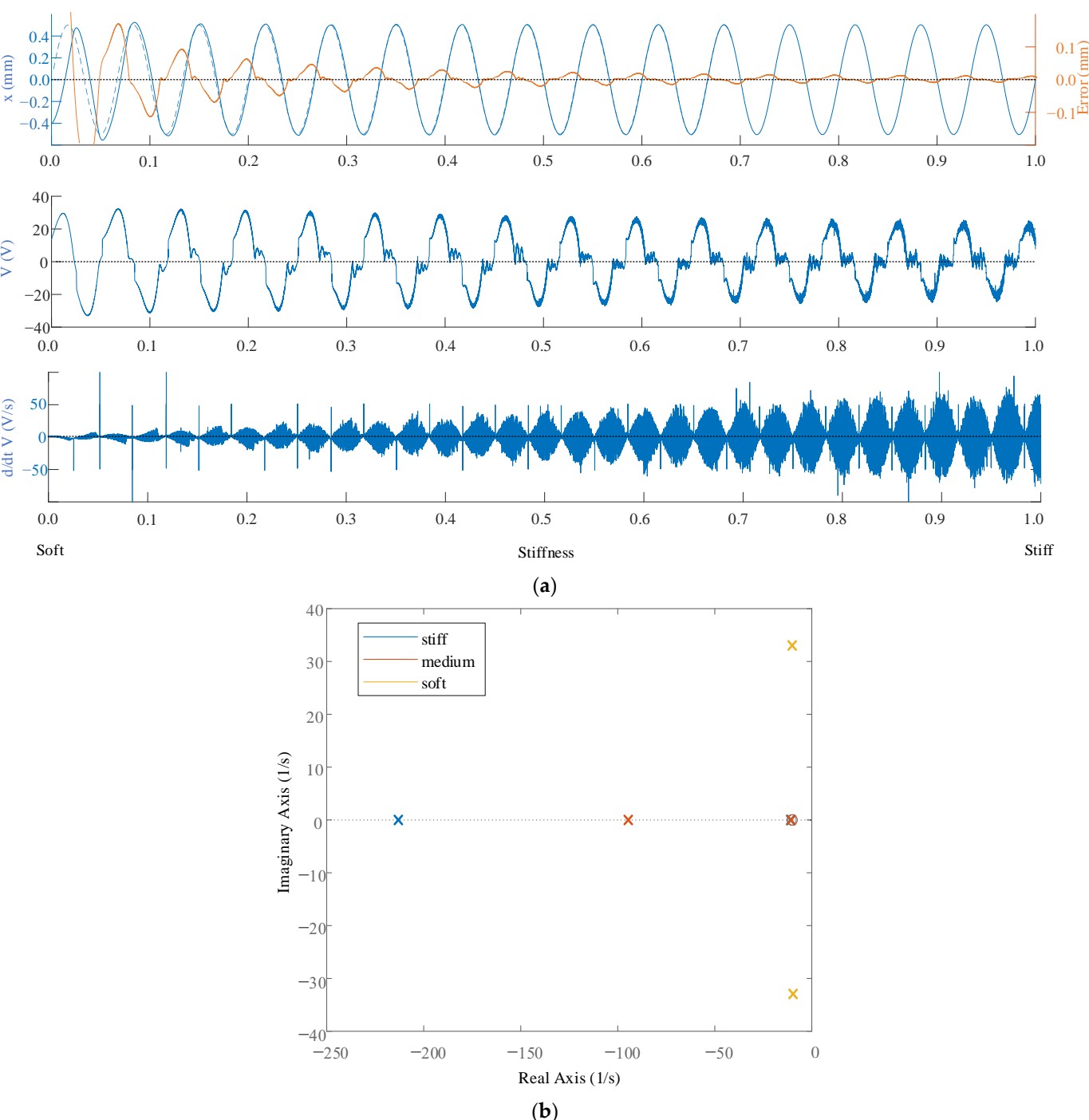

**Figure 5.** Effect of soft/stiff controller on the behavior of the closed−loop: experimental results.

The electro-mechanical linear stage can be controlled by an adaptive controller to suit a different type of motion. In our physical plant, the controller can be either an open loop with a programmable voltage or a closed loop with a programmable stiffness.

## 5. The Proposed Trajectory Controller Design with Digital Twin

Figure 6, the proposed trajectory controller is designed to have a digital twin framework. The digital twin is constructed with Matlab's Simscape. The mathematical model of motor, gear, ball screw, and overall friction are modeled in the digital twin. The reference trajectory is fed via both the physical controller and digital controller. The physical controller drives the physical twin, and the trajectory is measured by an encoder. Meanwhile, the digital controller drives the digital twin along. In this way, the real trajectory can be compared and analyzed with a predicted trajectory for anomaly detection. The digital twin provides an opportunity to try out different control strategies in the system to test its performance. Furthermore, the performance of a new controller can be analyzed. If there is a change in the physical plant or there is a disturbance, a comparison is made between the actual trajectory and predicted trajectory whereby the aggregated data is fed through a classifier to adjust the digital twin or switch to a suitable controller.

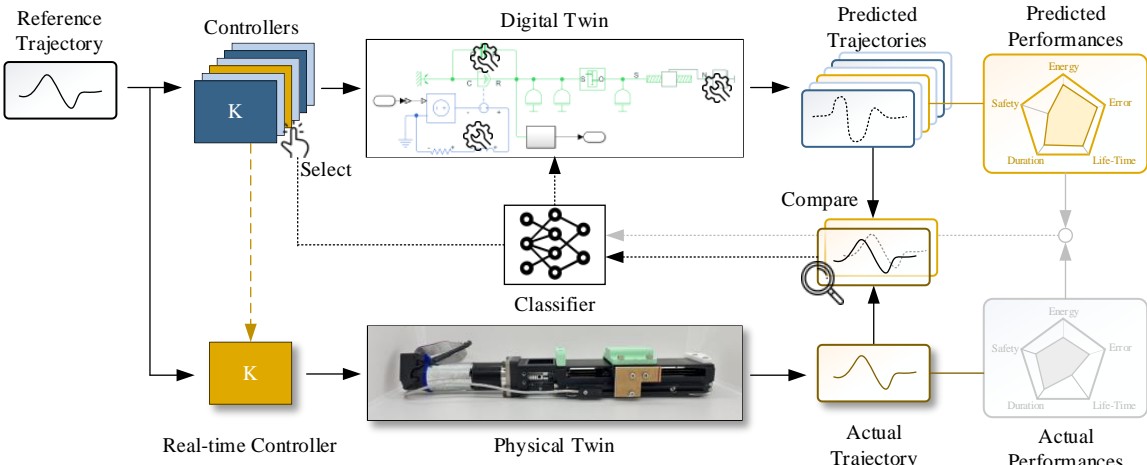

**Figure 6.** The proposed trajectory controller based on a digital twin framework.

In this work, the actual trajectory is compared with the predicted one in runtime. The joint controller is updated at 1 kHz while the comparison is carried out at 50 Hz. The difference is fed into the classifier. If the difference goes beyond the threshold, the classifier switches the controller to a suitable one and may update the applied parameters.

## 6. The Resulting Behavior of the Proposed Controller

The existing industrial linear stage is excellent for most applications but can be enhanced by having a digital twin framework. In this paper, anomaly detection and adaptive control are demonstrated.

### 6.1. Anomaly Detection

If there is unusual or unexpected behavior during operation, the resulting trajectory can differ from the expected one. Anomaly detection is normally used to detect potential problems or issues that can impact the performance or reliability of the linear stage. In our demonstration, the stiff controller follows a 1 Hz sinusoidal. At some point, we obstructed the ongoing motion. The tracking error image is obviously seen at one cycle time (Figure 7). When the obstruction was removed, the trajectory went back to normal. The projected behavior and/or the empirical historical data can be used as a normal reference. The empirical historical data is used in the experiment, as demonstrated in Figure 7.

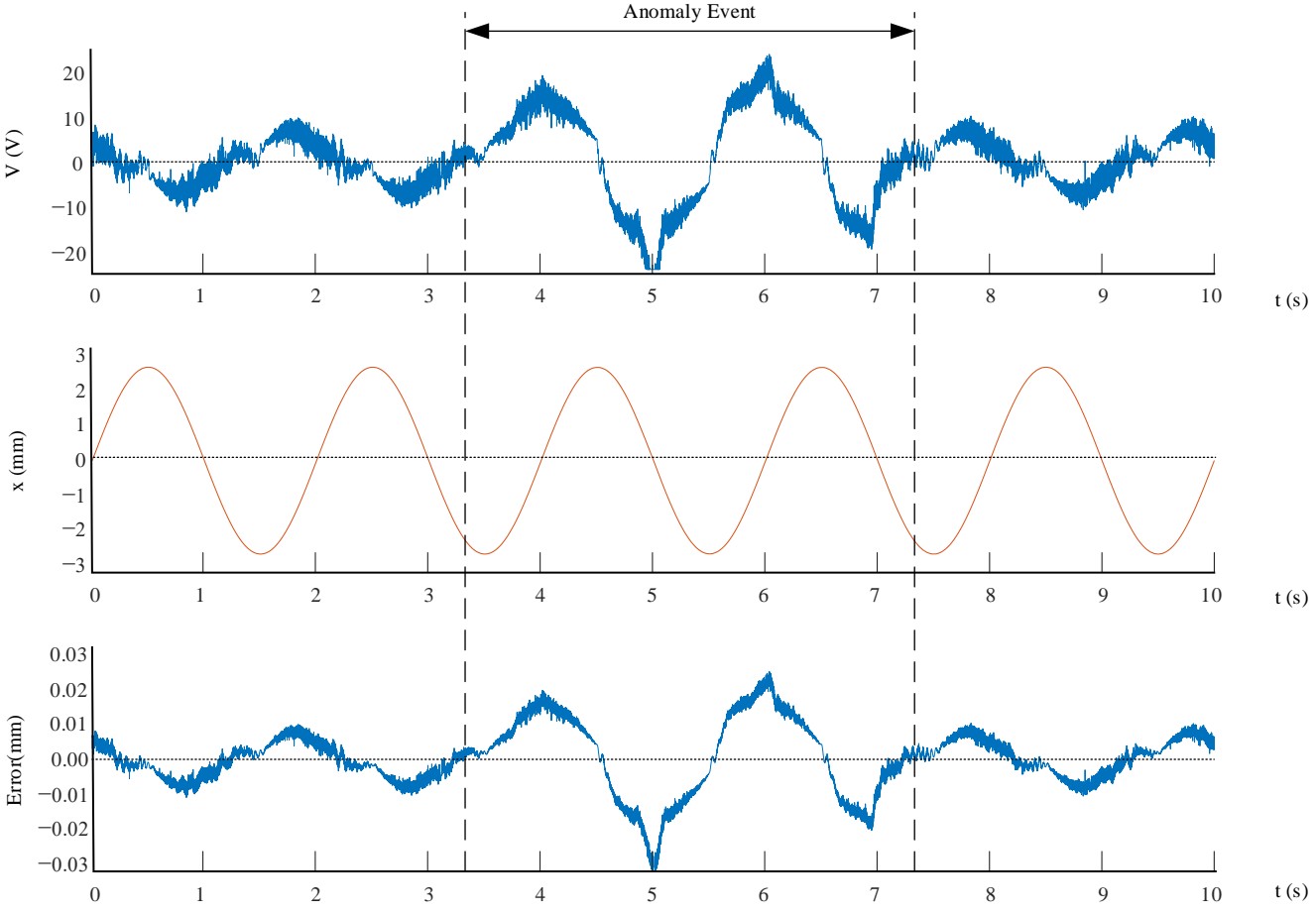

**Figure 7.** Anomaly detection.

*6.2. Controller Adaptability*

The next experiment is to demonstrate the adaptability of the proposed controller. Previously, detection of anomalies was carried out using an error threshold method, which compared errors found with empirical error data. The stiff controller followed a 0.5 Hz sinusoidal (Figure 8). At 3.3 s, we intentionally placed an obstruction in front of the carriage. Subsequently, the anomaly was successfully detected via the threshold technique, and the classifier switched the controller to the soft controller.

In this experiment, both anomaly detection and use of controllers: soft and stiff are demonstrated. During normal operation, the controller is stiff: such behavior is shown in Figure 8 before 3.3 s: observable maximum tracking error is 0.051 mm. In Table 3, afterhitting the obstacle at 3.3 s, the proposed anomaly detection detects this event with 0.095 s delay. The controller is switched to the soft controller where the observable maximum error is 0.535 mm. Subsequently, the effort continuously rose until it reached a maximum voltage of 24 V. When the controller switched from anomaly detection, the effort dropped and resulted in a peak voltage of 16.77 V.

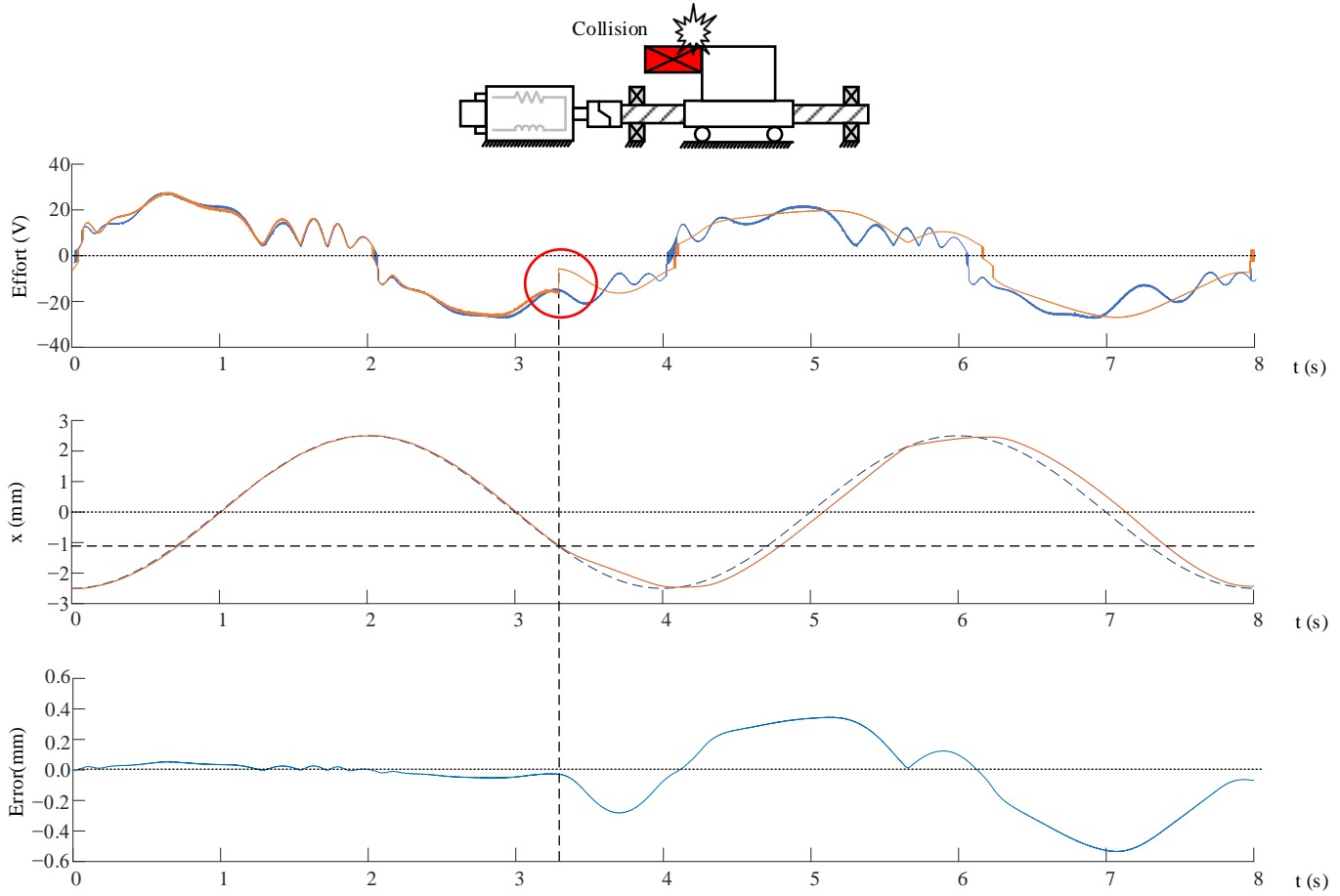

**Figure 8.** Anomaly detection and controller's adaptability.

**Table 3.** Observable performance of the proposed controller.

| Controller-Type | Maximum Error (mm) | Hitting Peak Voltage (V) |
|:---:|:---:|:---:|
| Stiff | 0.051 | 24 |
| Soft | 0.535 | 16.77 |

## 7. Conclusions

A trajectory controller having a digital twin framework for the electromechanical linear stage was successfully designed. Its capabilities are demonstrated. The controller can be programmed as an open loop control with a programmable voltage or a closed loop control with a programmable stiffness. The digital twin for the linear stage is successfully implemented via Matlab's Simscape along with a Beaglebone board. The digital twin runs along with the physical linear stage in real-time to equip the existing high precision controller with new capabilities. In this work, anomaly detection and adaptive soft-stiff controller are demonstrated. The controller was able to detect the anomaly, and consequently switched to soft controller; the voltage decreased to 69.86%, but the increment in maximum tracking error was 0.484 mm. In further work, it is proposed to use digital twin technology as a pathway for a novel system that we are researching, which incorporates an AI-embedded drilling and air supply for a pneumatic robot.

**Author Contributions:** Conceptualization, R.C.; methodology, K.C. and R.C.; software, K.C.; validation, K.C. and R.C.; formal analysis, K.C. and R.C.; investigation, R.C.; resources, R.C.; data curation, K.C.; writing—original draft preparation, K.C. and R.C.; writing—review and editing, K.C. and R.C.; visualization, K.C.; supervision, R.C.; project administration, R.C.; funding acquisition, R.C. All authors have read and agreed to the published version of the manuscript.

**Funding:** This project is funded by National Research Council of Thailand (NRCT).

**Data Availability Statement:** Not applicable.

**Conflicts of Interest:** The authors declare no conflict of interest. Furthermore, the funders had no role in the design of the study; in the collection, analysis, or interpretation of data; in the writing of the manuscript, or in the decision to publish the results.

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
