# Peer review of "A Deep Trajectory Controller for a Mechanical Linear Stage Using Digital Twin Concept"

_actuators, doi:10.3390/act12020091_

Round 1
Reviewer 1 Report
1. The abstract lacks the contribution.
2. Most the contribution points do not express a real contribution; but merely objectives. In addition, some points are not complete sentences like (The adaptive controller that can be either open loop with programmable voltage or closed loop with programmable stiff).
3. The literature has to be enriched by relevant work and I suggest the work to be added: https://doi.org/10.1080/21642583.2018.1547887
4. In section 2, is the device in Figure 2 a company-manufactured or manufactured by the authors. If it is company-manufactured, the authors have to supply the serial number.
5. The simscape model is not a breakthrough!!! The numerical simulation is simple to simulate the controlled system.
6. Table (1) is not important and the variables can be written within the text.
7. The modeling of the system is simple and it cannot represent complex and challenging application. In addition, there are linear motor motion which can replace this classical technology.
8. The author have to indicate which topology of DC configuration is used in modeling: Filed controlled or Armature controlled.
9. The authors have to explain how Eq.(3) has been obtained.
10. Figure (2) has to be explained in clearer manner.
11. Numerical Tables have to be established to show the effectiveness of the proposed controller.
12. The future work has to be added.
13. The conclusion is descriptive and percentages of improvement have to be numerically reported.
Author Response
First of all, I would like to express my gratitude for taking the time to review our work and providing valuable feedback.
We have taken your comments into consideration and have made the necessary revisions to the manuscript. We have taken them into account in our revisions, and we believe that they have greatly improved the manuscript. Please see the attachment.

Reviewer 2 Report
In the article under review, the authors propose a trajectory controller with its digital twin for an electromechanical linear stage. The studies were carried out using the MATLAB/Simscape software running in real-time with a Beaglebone black microcontroller board.
Undoubtedly, the studies conducted by the authors are relevant, the results of which can be useful to scientists and researchers, specialists in the field of control systems.
The paper provides a description of the electromechanical linear stage and its model implemented in Simscape. The proposed digital twin is described, including a mathematical model and simulation framework. Ideas for identifying the parameters of the digital twin are presented. A typical trajectory controller and the proposed trajectory controller design with digital twin are described. Some anormally detection and adaptive control are experimentally demonstrated.
However, during the review, I drew attention to the following shortcomings, the correction of which would improve the quality of the paper and I would also like to ask for clarification:
- In the Introduction section, the authors present a description of the industrial linear stage, and the advantages of using digital twins. However, it does not provide a review of the literature on the use of digital twins for trajectory controller and motion controllers. The Introduction should be substantially supplemented.
- Identification of the friction model was carried out experimentally by the authors. However, friction significantly depends on the temperature of the mechanical elements, which changes during operation. How, according to the authors, will this fact affect the error in predicting the trajectory of motion? Have the authors conducted research on this issue?
- Please specify in section 4 the results of simulation or experimental studies are given.
- It seems to me that in section 5 it is necessary to provide a more detailed description of the trajectory controller design with digital twin proposed by the authors with a detailed description of each block shown in Figure 6.
- Line 365 talks about enhancing the precision linear stage when using the digital twin paired with embedded artificial intelligence. However, the concept of "artificial intelligence" in this place in paper appears for the first time.
- Also, in my opinion, the results of experimental confirmation of the advantages of the proposed system with a digital twin are not presented in sufficient detail.
Author Response

(The authors gave the same response as above.)

Reviewer 3 Report
The work is interesting and within the scope of the journal. The authors should comment on any similarities and differences between the digital-twin approach as implemented here and established control methods such as Model Predictive Control. What are the advantages of their approach compared to other existing approaches?
The authors should provide details of the (neural) classifier and how it is trained. How the classifier adjusts the parameters of the system (digital twin?) should also be detailed.
The paper should be proofread again carefully and edited to remove the many grammatical mistakes found in the current manuscript.
Author Response

(The authors gave the same response as above.)

Round 2
Reviewer 1 Report
The authors have addressed most my comments. However, I hope the authors consider the following comments:
1. The comment 3 in previous report have not been considered.
2. The conclusion has not reflected the improvements numerically and decisively.
Author Response
Thank you for your comment. Please see the attachment.

Reviewer 2 Report
The authors provided answers to all my questions. In the new version of the paper, my recommendations were taken into account. In my opinion, the article in this version can be accepted.
Author Response

(The authors gave the same response as above.)
